# Concise Reasoning via Reinforcement Learning

## Abstract

A major drawback of reasoning models is their excessive token usage, inflating computational cost, resource demand, and latency. We show this verbosity stems not from deeper reasoning but from reinforcement learning loss minimization when models produce incorrect answers. With unsolvable problems dominating training, this effect compounds into a systematic tendency toward longer outputs. Through theoretical analysis of PPO and GRPO, we prove that incorrect answers inherently drive policies toward verbosity *even when* $\gamma = 1$, reframing response lengthening as an optimization artifact. We further uncover a consistent correlation between conciseness and correctness across reasoning and non-reasoning models. Building on these insights, we propose a two-phase RL procedure where a brief secondary stage, trained on a small set of solvable problems, significantly reduces response length while preserving or improving accuracy. Finally, we show that while GRPO shares properties with PPO, it exhibits collapse modes, limiting its reliability for concise reasoning. Our claims are supported by extensive experiments.

## 1 Introduction

Reasoning models have gained significant importance in both research and products, demonstrating remarkable performance in various domains. This success is largely attributed to extensive reinforcement learning (RL) (Sutton & Barto, 2018) post-training applied to a base model, which is initially trained through supervised learning for token completion. During RL training, the model is exposed to a diverse set of reasoning problems, along with their corresponding final answers. (Appendix A.1 discusses related work.) Notably, using a complete solution as the training target is not required in RL; instead, the model explores the response space, similar to how an RL agent learns to play a video game. This process fundamentally differs from the supervised training phase for human alignment, commonly referred to as reinforcement learning from human feedback (RLHF), where the objective is to select responses that align with human preferences among multiple model-generated alternatives.

A key phenomenon observed during RL post-training is the emergence of an "aha moment" (Guo et al., 2025). This refers to an inflection point where the model begins exhibiting self-correction behaviors, as seen in responses like "*We must have made a mistake, let's try again.*" Crucially, this behavior is not explicitly programmed but emerges naturally as the model explores the response space. Prior research has found a distinct pattern following this moment: response lengths tend to increase significantly, accompanied by improvements in overall accuracy (Guo et al., 2025; Yeo et al., 2025; Zeng et al., 2025). Even with a lack of clear understanding of why this happens, this phenomenon has led many to push for longer responses, leveraging additional training and computational resources in the hope of further enhancing accuracy.

We argue that the observed gains in accuracy are distinct from the tendency to generate longer responses. The push for lengthier outputs is not a reasoning strategy but a side effect of minimizing loss. This view also resolves the apparent paradox that, in both reasoning and non-reasoning models, correct responses to reasoning-intensive tasks are often shorter than incorrect ones.

Moreover, a critical but often overlooked observation is that RL post-training on moderate-size datasets from domains like mathematics often leads to an initial decrease in response length with no negative impact on the accuracy (Pan et al., 2025; Luo et al., 2025). This suggests that many tokens in the produced chain of thought may not be needed. Additionally, examining the chain of thought in reasoning models, such as DeepSeek's R1 (Guo et al., 2025), reveals a significant degree of

redundancy, repetition, and irrelevant artifacts. This observation raises a fundamental question: Can reasoning models be further optimized through RL training to produce systematically more concise chains of thought without sacrificing accuracy? To this end, and in light of our theoretical analysis, we propose a **two-phase RL training** paradigm designed to first enhance the reasoning capabilities of the base model and then enforce conciseness. Both phases can be trained on a small problem set, but the second is key to producing concise responses. Our approach offers a direct alternative to models such as DeepSeek's R1, yielding significantly shorter responses while maintaining accuracy and in some cases even improving it. It is cost-effective, requires minimal training, and improves computational efficiency, resource use, and response time.

Additionally, we present the following key findings.

1. **Conciseness and Accuracy Correlation:** We show that, during inference of both reasoning and non-reasoning models, *concise reasoning* strongly correlates with higher accuracy.

2. **Analysis of PPO Loss Function Dynamics:** We present a mathematical analysis establishing the link between response correctness and PPO's loss function. Specifically, we show that incorrect answers tend to drive longer responses, while correct ones encourage brevity.

3. **Analysis of GRPO Loss Function Dynamics:** We show that while GRPO encourages longer responses when facing negative advantage and promotes conciseness when facing positive advantage, it suffers from collapse modes, making it ineffective in enforcing conciseness.

4. **Limited Data:** We show that RL post-training phases are effective even with remarkably small datasets, a result that defies current trends in the literature and proves viable for resource-constrained scenarios, as confirmed by our experiments.

## 2 RESPONSE LENGTH VS. ACCURACY

A crucial yet often overlooked phenomenon is that in both reasoning and non-reasoning models, with or without RL training, brevity and accuracy are strongly correlated as shown in Table 1. Note that these results cover benchmarks with varying levels of difficulty. Therefore, the length-accuracy correlation is not an artifact of the problem difficulty. Moreover, as reported in previous works (Pan et al., 2025; Luo et al., 2025), RL post-training often produces significantly shorter responses, even without explicit penalties for length, while still preserving or improving correctness. This effect is particularly evident in the early stages of training.

| Benchmark | MATH500 | | AIME'24 | | MMLU-STEM | |
| Model | Correct | Incorrect | Correct | Incorrect | Correct | Incorrect |
|---|---|---|---|---|---|---|
| R1-Distill-Qwen-1.5B | $3518_{3314}$ | $11519_{686}$ | $7462_{72}$ | $14269_{168}$ | $1577_{9756}$ | $3431_{14388}$ |
| R1-Distill-Qwen-7B | $3204_{1858}$ | $10436_{142}$ | $6953_{64}$ | $14576_{56}$ | $818_{6121}$ | $1377_{5951}$ |
| Qwen2.5-Math-1.5B-Inst | $477_{5946}$ | $815_{2054}$ | $779_{51}$ | $989_{429}$ | $382_{27279}$ | $460_{21009}$ |
| Phi-4 | $529_{6397}$ | $1107_{1603}$ | $932_{80}$ | $1333_{400}$ | $383_{11417}$ | $406_{36871}$ |

Table 1: Average response length for correct and incorrect answers across different models and benchmarks. The blue subscripts indicate the number of samples used to compute the averages.

These observations are in sharp contrast with the widespread belief that very long chains of thought are inherently necessary to achieve higher accuracy (Guo et al., 2025; Yeo et al., 2025; Zeng et al., 2025). A related explanation for reduced accuracy in long responses is the notion of deadends, states where the correct answer becomes unlikely (Fatemi et al., 2019; 2021; Cao et al., 2023).

Observing that long responses are not necessarily correlated with accuracy, a key question remains: When and why do LLMs, trained with RL, tend to increase response length? To answer this question, we return to RL fundamentals, formalizing each reasoning problem as an MDP in the next section.

## 3 EACH REASONING PROBLEM IS AN MDP

Each reasoning problem (e.g., a math problem) fundamentally constitutes a Markov Decision Process (MDP) (Puterman, 2014) rather than a mere static sample. An MDP consists of a state space $\mathcal{S}$, an

action space $\mathcal{A}$, a transition function $T$, a reward function $R$, an initial state distribution $\mathcal{P}_0$, and a discount factor $\gamma$. In language modeling, the state at each token position $k$ consists of all tokens (or their embeddings) up to and including $k$, as well as any contextual information such as the problem statement. The action space corresponds to the vocabulary of possible tokens. The transition function deterministically appends new tokens to the sequence. The reward function is zero for all steps except the final one, where correctness is evaluated based on final answer and formatting. The initial state depends on the prompt, which may include problem statements and directives (e.g., "Solve step by step and enclose the final answer in a box."). The RL objective is to maximize the expected return, referred to as the value function and denoted by $V(s)$, where return is defined as the sum of future rewards discounted by $\gamma$. It is common practice in LLM post-training to set $\gamma$ to 1.

We measure problem difficulty for LLMs by the proportion of correct responses, since sampling with non-zero temperature yields difference responses. Let $p_a$ be the probability a problem is solved in at least one attempt: $p_a > 0$ means occasionally solvable, $p_a = 1$ fully solvable, and $p_a = 0$ unsolvable. For example, for R1-Distill-Qwen-1.5B, 21/30 AMIE24 and 464/500 MATH problems were occasionally solvable. Solving a problem when only the final answer is given requires a base model capable of occasional correct solutions, akin to an agent playing an interactive game.

When training on multiple problems, the overall MDP consists of multiple initial states with an updated reward function. Adding more problems modifies $P_0$ and $R$ but retains the fundamental MDP structure. This introduces two considerations: (1) A larger set of problems increases the MDP complexity, which may lead to greater generalization of the learned skills. (2) In principle, a small set of problems (even a single one) should be sufficient for RL training to take effect, though this may raise concerns about overfitting.

Overfitting is a challenge in supervised learning, where models tend to memorize training data instead of generalizing. Online RL, by contrast, continuously generates new responses, allowing the model to refine its reasoning over time. Rather than imitating fixed solutions, it actively explores and reinforces successful reasoning strategies. Two factors make online RL robust: (1) sampling techniques like non-zero temperature encourage response diversity, and (2) continual model updates shift response distributions, reducing stagnation. This enables RL to remain effective even with an extremely small set of training problems; a novel aspect of this work not previously explored in the literature.

## 4 PPO IMPACT ON RESPONSE LENGTH

To study how response length relates to RL loss, we trained DeepSeek-R1-Distill-Qwen-1.5B using Proximal Policy Optimization (PPO) (Schulman et al., 2017) on only 4 OlympiadBench (He et al., 2024) problems (see Appendix A.4). These problems were specifically chosen because the base model consistently failed to solve them despite extensive sampling, yielding **a constant terminal reward of $-0.5$**. The context size was 20K tokens. Fig. 1 shows policy loss against response length, revealing a strong correlation: as length increased, loss consistently decreased. This clearly indicates longer responses do not reflect improved reasoning, as the model still fails entirely; rather, they arise from loss dynamics under negative reward.

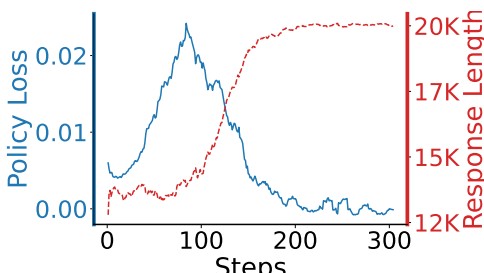

Figure 1: **Effect of loss on response length**, smoothed with a 10-step moving average.

Using generalized advantage estimation (GAE) (Schulman et al., 2016), the advantage function is estimated as a $\lambda$-weighted sum of step-wise TD-errors, akin to forward-view eligibility traces in value estimation (Sutton & Barto, 2018; Bertsekas & Tsitsiklis, 1996). PPO then uses GAE to estimate both the policy and the value losses. In practice, PPO is often used with $\gamma = 1$ and $\lambda = 0.95$.

Consider a response of length $T$ (with token indices from 0 to $T - 1$), where a terminal reward $r \neq 0$ is applied only at the final step, $T - 1$. Let $\gamma = 1$. In PPO, per-token loss is averaged over the full response length $T$. We assume $|V(s_{t+1}) - V(s_t)| \leq \epsilon$ for all $t$; this is expected since the value network's output space is bounded and Lipschitz. We show that when $\lambda < 1$, PPO favors shorter

responses if $r > 0$ and longer responses if $r < 0$. Importantly, setting $\lambda = 1$ may introduce an undesirable bias toward shorter responses regardless of correctness and significantly amplifies noise, with noise scaling linearly in response length.

By design, $r_t = 0$ for $t < T - 1$, and $r_t = r$ for $t = T - 1$. With the temporal-difference (TD) error $\delta_t = r_t + V(s_{t+1}) - V(s_t)$ and $\gamma = 1$, the GAE for token $t$ is $\hat{A}_t = \sum_{l=0}^{T-t-1} \lambda^l \delta_{t+l}$, and the PPO per-token loss is given by $L_t := -\min\big(\rho_t \hat{A}_t, \text{clip}(\rho_t, 1 - \epsilon_{\text{clip}}, 1 + \epsilon_{\text{clip}}) \hat{A}_t\big)$, where $\rho_t = \rho_t(\theta, \theta_{old}) := \frac{\pi_\theta(o_t | o_{<t})}{\pi_{\theta_{\text{old}}}(o_t | o_{<t})}$ is the importance sampling ratio. It yields:

$$L_t = -\alpha_t \hat{A}_t, \qquad \alpha_t := \begin{cases} \min\big(\rho_t, \text{clip}(\rho_t, 1 - \epsilon_{\text{clip}}, 1 + \epsilon_{\text{clip}})\big), & \hat{A}_t > 0, \\ \max\big(\rho_t, \text{clip}(\rho_t, 1 - \epsilon_{\text{clip}}, 1 + \epsilon_{\text{clip}})\big), & \hat{A}_t < 0, \end{cases}$$

We assume $0 \leq \rho_{\min} \leq \rho_t \leq \rho_{\max} < \infty$ (this is a fair assumption, due to regularization). The complete PPO loss $L$ and the *unweighted mean advantage* $S$ are defined as

$$L := \frac{1}{T} \sum_{t=0}^{T-1} L_t = -\frac{1}{T} \sum_{t=0}^{T-1} \alpha_t \hat{A}_t, \quad S := -\frac{1}{T} \sum_{t=0}^{T-1} \hat{A}_t$$

The following result characterizes the asymptotic behavior of the PPO loss under GAE:

**Theorem 1 [PPO Loss under GAE]:** Let $r_t = 0$ for $t < T - 1$, $r_{T-1} = r$, and $V(s_T) = 0$. Define the TD errors $\delta_t := r_t + V(s_{t+1}) - V(s_t)$ and set $R := r - V(s_{T-1})$. Assume $\text{sgn}(R) = \text{sgn}(r)$ and $|\delta_k| \leq \epsilon$ for $k < T - 1$. Then,

$$S = \begin{cases} -\dfrac{R}{T(1-\lambda)} + O\big(\dfrac{\epsilon}{1-\lambda}\big), & \lambda < 1 \\ -R + O\big(\dfrac{\epsilon T}{2}\big), & \lambda = 1 \end{cases} \tag{1}$$

Moreover, if $0 \leq \rho_{\min} \leq \rho_t \leq \rho_{\max} < \infty$, and $\alpha_{\text{dev}} := \max\{1 - \rho_{\min}, \rho_{\max} - 1, \epsilon_{\text{clip}}\}$, then $\big|L - S\big| \leq \alpha_{\text{dev}} \frac{1}{T} \sum_{t=0}^{T-1} |\hat{A}_t|$. Thus, $L$'s behaviour closely follows that of $S$ to the level of this bound. If, in addition, all $\hat{A}_t$ share the same sign $\sigma \in \{\pm 1\}$, then $\text{sgn}\, L = \text{sgn}\, S$ and

$$\sigma = +1: \quad \rho_{\min} |S| \leq |L| \leq \min(\rho_{\max}, 1 + \epsilon_{\text{clip}}) |S|,$$
$$\sigma = -1: \quad (1 - \epsilon_{\text{clip}}) |S| \leq |L| \leq \rho_{\max} |S|.$$

**Remark 1.** In practice, $\hat{A}_t$ should be bounded (otherwise, loss diverges). Thus, $\frac{\alpha_{\text{dev}}}{T} \sum_{t=0}^{T-1} |\hat{A}_t| \leq C\alpha_{\text{dev}}$ for some $C > 0$, and the behavioral similarity between $S$ and $L$ is even more pronounced.

**Interpretations.** The theorem leads to the following key points:

- **For $\lambda < 1$:**
  - The average error term is bounded by a constant, $O\big(\epsilon/(1-\lambda)\big)$, independent of $T$.
  - When the terminal term dominates the pre-terminal error, the sign is determined by $-R$: if $R < 0$ then $L > 0$ and its magnitude decreases like $1/T$ (*longer responses are favored*); if $R > 0$ then $L < 0$ and its magnitude increases as $T$ decreases (*shorter responses are favored*).
  - Extensive experiments confirm that consistently negative rewards yield positive loss (e.g., Fig. 1), while predominantly positive rewards yield negative loss.

- **For $\lambda = 1$:**
  - The average error scales linearly with $T$, making PPO highly sensitive to value estimation errors, leading to significantly slower, inefficient, or unstable training (see Appendix Fig. 7).
  - Enforcing long responses can be difficult even when answers are consistently incorrect.
  - ***Return*** is calculated by $\hat{A}_t + V_t^{old} = r - V_{T-1}^{old} + V_t^{old} + \sum_{l=1}^{T-t-2} \delta_{t+l}$ and is used as *target* for training $V_t$. When $r < 0$, the positive $V_t^{old} - V_{T-1}^{old}$ pushes the target upward, causing overflow, especially for early tokens. When $r > 0$, it can pull the target downward, leading to underflow (see Appendix Figs. 6 and 7). Notably, when $\lambda < 1$, return is discounted, preventing this effect.

**Remark 2.** For $\gamma < 1$, including $\gamma$ in the analysis has two effects: (1) similar to $\lambda$, it directly affects the loss, and (2) it modifies the TD errors, introducing a lower bound on $\epsilon$ that may degrade training performance. Thus, we recommend using $\lambda$ instead (see Appendix A.2 for more details).

## 5 GRPO IMPACT ON RESPONSE LENGTH

In the case of GRPO (Shao et al., 2024), loss is calculated from the following formula (note the negative sign):

$$L_{\text{GRPO}}(\pi_\theta) = -\mathbb{E} \frac{1}{G} \sum_{i=0}^{G-1} \frac{1}{|o^{(i)}|} \sum_{t=0}^{|o^{(i)}|-1} \min \left[ \rho_t^{(i)} \hat{A}_t^{(i)}, \text{clip}\left(\rho_t^{(i)}, 1-\epsilon, 1+\epsilon\right) \hat{A}_t^{(i)} \right] \quad (2)$$

where the expectaion is taken over $q \sim p_Q$, and $\{o^{(i)}\}_{i=1}^G \sim \pi_{\theta_{\text{old}}}(\cdot|q)$, with $\pi_\theta$ and $\pi_{\text{old}}$ are the main policy and the one before the update, $\rho_t^{(i)}$ is the IS ratio of the $i$–th response. Advantage is given by

$$\hat{A}_t^{(i)} = \frac{r(q, o^{(i)}) - \text{mean}(\{r(q, o^{(1)}), \dots, r(q, o^{(G)})\})}{\text{std}(\{r(q, o^{(1)}), \dots, r(q, o^{(G)})\})} \quad (3)$$

where $o^{(i)}$ and $o_t^{(i)}$ denote the $i$–th response and the $t$–th token in the $i$–th response, respectively. Liu et al. (2025) argued that the presence of $|o^{(i)}|$ in the denominator biases the model by encouraging longer responses when the advantage is negative and vice versa. To address this perceived issue, they proposed removing $|o^{(i)}|$ from the denominator. Unfortunately, this interpretation misses important details. As response length increases, both the numerator and the denominator grow, not just the denominator. Thus, the relationship between response length and loss is more nuanced than they imply. We present an alternative argument showing that the GRPO loss encourages longer responses for negative advantages and shorter responses for positive ones. We also show that GRPO has notable shortcomings, casting doubt on its effectiveness, particularly for improving conciseness.

Two key observations: (1) by definition, advantage is independent of token position $t$, so the subscript is redundant and can be moved outside the inner sum; (2) although GRPO typically uses non-negative rewards, $\hat{A}$ reverses their effect. With some responses labeled positive and others zero, the mean in equation 3 lies between 0 and 1, yielding $\hat{A} > 0$ for correct responses and $\hat{A} < 0$ for incorrect ones. This also highlights another similarity between GRPO and PPO's common practice, where incorrect responses incur negative rewards. The following theorem summarizes GRPO's core behaviour.

**Theorem 2 [GRPO Prolixity]:** Let $\pi_\theta$ and $\pi_{\theta_{\text{old}}}$ be the current and old policies in GRPO training. Consider a sampled response ending in a terminal token $\tau$, with advantage $\hat{A}$. Then, (1) if $\hat{A} < 0$, decreasing $\pi_\theta(\tau)$ reduces the loss; and (2) if $\hat{A} > 0$, increasing $\pi_\theta(\tau)$ reduces the loss.

This result shows that $\hat{A} < 0$ is a sufficient condition for encouraging longer responses, a phenomenon widely observed by practitioners. When $\hat{A} > 0$, it implies a higher chance of termination at the current position once updated, but it does not favor shorter responses among multiple correct completions. To address this, we present a general result, mainly relevant to GRPO, focusing on what happens when *two correct responses differ in length*. Without loss of generality, we assume the first token of each response acts as the trigger guiding the model toward that sequence. To simplify notations, we remove conditioning from policies. From equation 2, for any *correct* response $i$, we have

$$L^{(i)} = -\hat{A}^{(i)} \cdot \frac{1}{|o^{(i)}|} \sum_{t=0}^{|o^{(i)}|-1} \min\left(\rho_t^{(i)}, 1+\epsilon\right), \quad \rho_t^{(i)} = \frac{\pi_\theta(o_t^{(i)})}{\pi_{\theta_{\text{old}}}(o_t^{(i)})}$$

**Theorem 3 [General Conciseness]:** Let $\pi_\theta$ be a "*softmax*" policy over a vocabulary of size $K$. Let $o_0^{(S)}$ and $o_0^{(L)}$ denote the first tokens of two responses generated by $\pi_{\theta_{\text{old}}}$ leading to sequence lengths $T_S$ and $T_L > T_S$, respectively. Assume: (1) both responses share the same positive advantage $\hat{A} > 0$; and (2) the importance sampling ratios satisfy $\rho_0^{(S)}, \rho_0^{(L)} < 1 + \epsilon$. Define: $f(k) := \left[ \left(1 - \pi_\theta(o_t^{(i)} = k)\right)^2 + \sum_{j \neq k} \pi_\theta(o_t^{(i)} = j)^2 \right]^{\frac{1}{2}}$. Then,

1. For the $k$–th token of vocabulary, the gradient norm induced by the softmax layer w.r.t. the logits $\boldsymbol{z}$ is $f(k)$.

2. $\left\| \nabla_\theta L^{(S)}(o_0^{(S)}) \right\| > \left\| \nabla_\theta L^{(L)}(o_0^{(L)}) \right\|$ iff $\rho_0^{(S)} f(o_0^{(S)}) > \frac{T_S}{T_L} \tilde{\kappa} \rho_0^{(L)} f(o_0^{(L)})$, with $\tilde{\kappa} > 0$.

3. This result is invariant under softmax temperature rescaling.

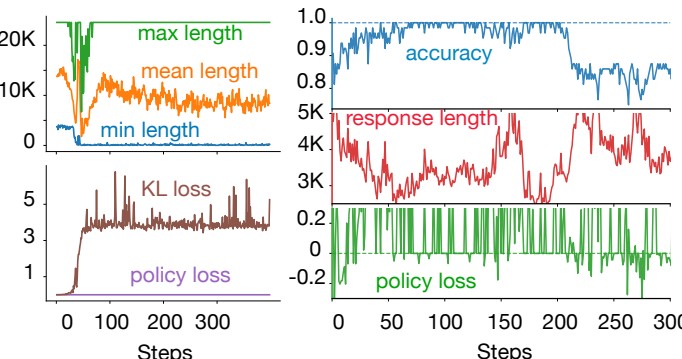

Figure 2: **GRPO on unsolvable and easy problems. Left**: On unsolvable problems, both policy loss and reward remained zero and the KL resulted in the collapse of **min** response length. **Right**: On easy problems, the policy loss dropped to zero when accuracy hit one, disrupting the learning process.

**Interpretations.**

- Under similar conditions (i.e., when $\pi_\theta(o_0^{(S)}) = \pi_\theta(o_0^{(L)})$ and $\rho_0^{(S)} = \rho_0^{(L)}$), the inequality generally holds because $T_S/T_L < 1$; hence, the shorter response receives stronger reinforcement (note that $L < 0$ for both responses since $\tilde{A} > 0$).

- The update almost surely favors shorter responses when $T_L \gg T_S$, or $\rho_0^{(S)} f(o_0^{(S)}) \gg \rho_0^{(L)} f(o_0^{(L)})$.

- If either $\rho_0^{(S)}$ or $\rho_0^{(L)}$ hits $1 + \epsilon$, the gradient is clipped to zero, preventing further preference.

- The possible range for $\tilde{\kappa}$ depends on the Jacobian's singular values, but experimental results indicate that $\tilde{\kappa}$ has a minor effect on the inequality, as conciseness still occurs under GRPO.

The Collapse of GRPO

In Appendix A.5, we show that the advantage function in GRPO admits the following closed form:

$$\hat{A} = \frac{r - \mu}{\sigma} = \begin{cases} \sqrt{(N-k)/k}, & \text{if } r = 1 \\ -\sqrt{k/(N-k)}, & \text{if } r = 0 \end{cases} \quad (4)$$

where $k$ denotes the number of correct responses within a group of size $N$. Notably, $\hat{A}$ is invariant under reward scaling. Expression (4) induces $\hat{A} = 0$ whenever the group is entirely correct or entirely incorrect. This structural property has critical implications for the learning dynamics of GRPO.

**First, unsolvable problems:** When problems are unsolvable, GRPO behaves differently from PPO. Since $\hat{A} = 0$, Theorem 2 does not apply and the loss is dominated by the KL term, which discourages deviation from the base model. To explore this further, we trained GRPO on four OlympiadBench problems using DeepSeek-R1-Distill-Qwen-1.5B (Fig. 2, left). Surprisingly, the model quickly converged to extremely short outputs (fewer than 80 tokens), likely because shorter responses incur lower KL penalties than longer ones.

**Second, fully solvable problems:** A similar failure mode arises when problems are consistently solvable. As accuracy hits 1, the estimated advantage $\hat{A}$ converges to zero, causing the policy loss to vanish. In this regime, the KL divergence term once again dominates the loss. We observed this effect by training GRPO on eight problems from the MATH dataset (Fig. 2, right). Once the model consistently solved all examples, progress stalled. In large datasets, this issue is often masked, as training batches are likely to include at least some partially solved or unsolved examples with non-zero advantage. However, in smaller or skewed datasets dominated by fully solvable problems, the advantage collapses across the board, reactivating the KL penalty as the primary learning signal. As a result, Theorem 3 becomes inapplicable, and GRPO loses its conciseness-inducing effect. While GRPO typically outperforms PPO in early training, this advantage can erode over time unless the KL term is carefully managed or adaptively reweighted.

Crucially, PPO does not suffer from these issues. In PPO, response length directly affects the advantage, whereas in GRPO, the advantage is fixed. This difference alone makes PPO more effective at promoting conciseness. Combined with the limitations noted above, these factors support using PPO over GRPO as a more robust method, especially when conciseness is a key objective.

## 6 A TWO-PHASE REINFORCEMENT LEARNING STRATEGY

Our analysis highlights key dynamics in response length during RL training. When models are trained on unsolvable problems, response length tends to increase, as longer outputs are more likely to reduce the loss amid predominantly negative rewards. Conversely, for occasionally solvable problems, response length typically decreases. In large-scale settings, this behavior becomes complex and closely tied to problem difficulty. As more problems become at least occasionally solvable, we expect average response length to eventually decrease.

In practice, RL training data often contains difficult problems. As long as some problems consistently yield negative rewards, the tendency toward longer responses can persist, especially early in training or once easier problems are mastered. This push towards verbosity, combined with LLMs' strength in producing coherent text, may contribute to the "aha moment," producing elaborate but not necessarily more accurate answers. It is also important to note that longer responses do not imply persistent failure. The model may simply stumble on the correct answer due to its long chain of thought. When this happens, RL reinforces success regardless of token count, which explains why accuracy can improve even amid growing verbosity. The key point is that verbosity is a consequence of RL minimizing its loss in the face of negative rewards, and it is this verbosity that may occasionally lead to improved accuracy, which then would be reinforced, not the other way around.

If the dataset contains an excessive number of unsolvable problems, the transition from promoting longer responses to encouraging conciseness can be significantly delayed and costly. To overcome this, we propose a novel approach: enforcing conciseness through a subsequent phase of RL training with a dataset of occasionally solvable problems. This structure introduces a two-phase RL training:

1. In the first phase, the model is trained on challenging problems. This phase aims to enhance the model's problem-solving capacity, with an expected increase in response length as PPO/GRPO mostly encounters wrong answers, driving the model toward longer responses. Notably, this first phase can also be seen as the RL training of off-the-shelf reasoning models.

2. In the second phase, training continues on problems with non-zero $p_a$ (occasionally solvable). This phase enforces conciseness while preserving or even enhancing accuracy. Notably, as we will see, it also substantially improves the model's robustness to lowering the temperature, ensuring remarkable performance even with limited sampling.

A critical insight from the MDP perspective is that effective RL training can be achieved even with a small problem set, though at the cost of possibly reduced generalization. In particular, in the second phase of training, where the model has already developed generalization capabilities, PPO can be applied to a minimal dataset consisting of only a few problems.

## 7 EXPERIMENTAL RESULTS

In our experiments, we broadly show that our two-phase reinforcement learning strategy leads to significant improvements across various models. We begin by examining the impact of problem difficulty level ($p_a$) and demonstrate how RL can influence response length depending on the difficulty of training problems. Next, we demonstrate that a second phase of training on R1 models, using only eight problems, achieves significantly more concise reasoning across various benchmarks, while preserving (even improving) accuracy. Additionally, this second phase of RL post-training substantially enhances model robustness under reduced sampling intensity. Finally, revisiting the first phase, we establish that *non-reasoning models* can be vastly improved through minimal-scale RL training. These results highlight the broad applicability and efficiency of our approach.

We used DeepSeek-R1 models distilled on Qwen models as our base and applied RL post-training to the 1.5B and 7B variants. During each training cycle, the model generated eight independent responses per training example, which were scored based on their format and the correctness of the final answer. For PPO, we used the following reward scheme: +1 if the final answer was correct and enclosed in a box; –0.5 if the answer was boxed but incorrect; and –1 if no boxed answer was provided. For GRPO, we used a simpler reward scheme: +1 for a correct and boxed final answer, and 0 otherwise. (See Appendix A.8 for more information about the experimental setup.)

## 7.1 IMPACT OF PROBLEM DIFFICULTY ON ACCURACY-LENGTH CORRELATION

In this section, we present experimental results supporting our claim that RL training on occasionally solvable problems leads to shorter response lengths. This reduction correlates with problem difficulty and an increased probability of arriving at a correct answer ($p_a$), which in turn promotes more concise responses.

We considered 3 sets of training examples, each containing 4 problems from the AIME'24 dataset. To estimate problem difficulty, we evaluated the base model using 64 samples per problem and temperature $0.6$. The average $p_a$ for the three sets, from easiest to hardest, was 0.375, 0.25, and 0.16, respectively. Fig. 3 shows the accuracy and response length over training steps for both cases of PPO and GRPO. Across all problem sets and algorithms, improvements in accuracy coincided with reductions in response length, indicating that as the model became more accurate, its responses became shorter.

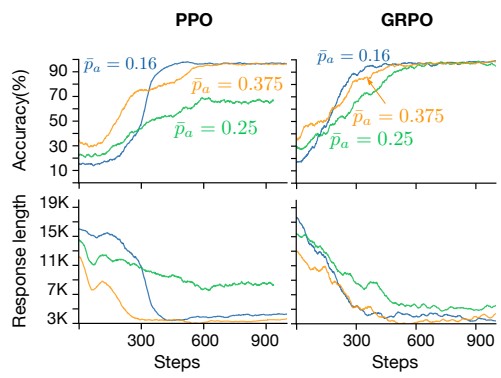

Figure 3: **Impact of difficulty levels**, smoothed with a 50-step moving average. Accuracy improvements consistently aligned with shorter response lengths across problem sets and training algorithms.

## 7.2 DECREASE IN RESPONSE LENGTH

In this section, we show the effect of RL post-training on eight examples randomly picked from the training subset of MATH dataset (Hendrycks et al., 2021) on reducing response length. We report PPO results, as GRPO performs unreliably on easy problems due to the collapse discussed in Section 5. (See Appendix A.12 for GRPO results). Although training was conducted exclusively on examples from the MATH dataset, we evaluated the models on AIME 2024 and AMC 2023 as well. For evaluation, we generated four samples per query with temperature of $0.6$ and top-p of $0.95$. Fig. 4 (left) shows the accuracy and response length of the post-trained models over training steps, evaluated on *test* datasets of AIME 2024, AMC 2023, and MATH-500: the response length decreased significantly, while accuracy remained stable across benchmarks and model sizes. To investigate whether reducing response length through RL post-training generalizes to other domains, we evaluated the 1.5B and 7B models trained on the eight training examples from the training subset of MATH dataset using the MMLU benchmark. The MMLU benchmark includes multiple-choice questions across 57 diverse subjects. For this evaluation, we specifically used MMLU-STEM, which focuses on science and engineering subjects such as physics, biology, and computer science. This subset contains 3018 distinct problems. The results are illustrated in Fig. 4 (right). As with the math domains, PPO post-training led to shorter response lengths on MMLU. Surprisingly, even with training on just eight examples, RL post-training also resulted in an accuracy improvement.

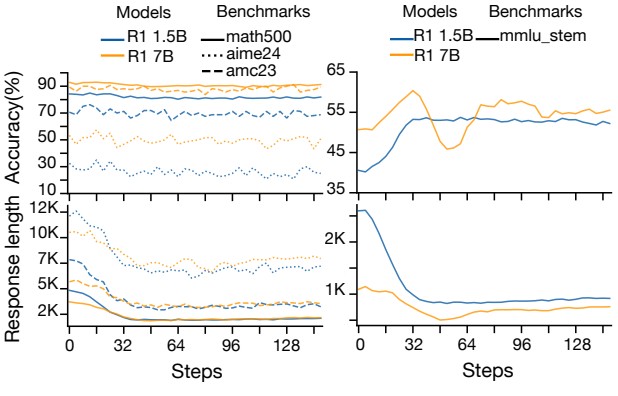

Figure 4: Response dynamics of two base models, R1-1.5B and R1-7B. Both models are trained with PPO using 8 problems from the level-5 subset of MATH dataset. **Left:** three mathematics benchmarks (different line styles). **Right:** *STEM* subset of the MMLU dataset. Response length and accuracy are shown against training checkpoints (steps). Response length decreased significantly, while accuracy remained stable or improved across benchmarks and model sizes.

Table 2 summarizes models' performance on 8 problems from the MATH training subset. Checkpoints were selected to minimize response length while maintaining reasonable accuracy.

Table 2: Comparison of R1 1.5B and R1 7B, and their post-trained versions on various benchmarks.

| Benchmarks | R1 1.5B | | | | R1 7B | | | |
| | Accuracy (%) | | Length (tokens) | | Accuracy (%) | | Length (tokens) | |
| | Baseline | Ours | Baseline | Ours | Baseline | Ours | Baseline | Ours |
|---|---|---|---|---|---|---|---|---|
| math500 (500) | 84.2 | 81.0 | 4842 | 1965 | 92.9 | 90.3 | 3718 | 2041 |
| aime24 (30) | 32.5 | 30.0 | 12104 | 6752 | 53.3 | 51.7 | 10510 | 6632 |
| amc23 (40) | 70.6 | 69.4 | 7847 | 2936 | 89.4 | 88.1 | 5674 | 3220 |
| mmlu_stem (3018) | 40.6 | 53.1 | 2597 | 821 | 50.7 | 58.1 | 1093 | 701 |
| average | 57.0 | **58.4** | 6848 | **3119** | 71.6 | **72.1** | 5249 | **3149** |

## 7.3 INCREASE IN PERFORMANCE AND ROBUSTNESS

This section shows that further RL post training also improves the model in terms of robustness and performance. To evaluate robustness, we examined sensitivity to temperature settings. Setting the temperature to zero can severely degrade the accuracy of reasoning models like R1. However, standard metrics such as pass@1, which rely on multiple samples at non-zero temperatures, often obscure the benefits of secondary RL post-training on a small dataset. We experimented with temperature 0 and 0.6 and observed that at temperature 0, the post trained model significantly outperformed the baseline model, suggesting the robustness of the post trained model compared to the base model (see Table 3).

| | Base Model | | Ours | |
| | MATH500 | AIME24 | MATH500 | AIME24 |
|---|---|---|---|---|
| $\tau$=**0.**, $n$=1 | 70% | 13.3% | 81% | 23.3% |
| $\tau$=**0.6**, $n$=4 | 84.3% | 32.5% | 81% | 30% |
| Rel. Degrade | **16.9%** | **59%** | **0%** | **22.3%** |

Table 3: Performance degrade with temperature $\tau$.

We also show that limited RL training on just a few examples can substantially boost accuracy, though the effect depends on prior RL exposure. When a model has already undergone extensive RL training, further gains are harder to achieve. To test this, we applied online RL to Qwen-Math-v2.5 on only four MATH examples, randomly picked from those with small but nonzero $p_a$. Unlike R1, this model had been trained solely with token completion. As shown in Table 4, we observed a surprisingly large improvement—up to 30% in the 1.5B model—demonstrating that even minimal RL post-training can yield major gains, particularly for models without earlier RL-based refinement.

| Model | MATH500 | | AIME24 | | AMC23 | |
| | Base | Ours | Base | Ours | Base | Ours |
|---|---|---|---|---|---|---|
| Qwen2.5-Math-7B | 47.5 | 67.1 | 15.8 | 23.3 | 43.8 | 57.5 |
| Qwen2.5-Math-1.5B | 33.5 | 63.1 | 6.7 | 9.2 | 30.0 | 48.1 |
| Qwen2.5-7B | 54.3 | 70.2 | 5.0 | 10.8 | 40.0 | 54.4 |

Table 4: Comparison of the base model and the RL-trained model using 4 examples (ours) across different base models and benchmarks shows a substantial accuracy gain with RL training.

## 8 CONCLUDING REMARKS

We presented a comprehensive view of how response length interacts with performance in reasoning models. We proposed a two-phase RL post-training strategy to first improve reasoning in the base model followed by enforcing conciseness. Our methodology substantially improved R1 models by achieving **over 54% and 40% reduction in response length for the R1 1.5B and R1 7B models**, respectively, while preserving accuracy and delivering significantly improved performance at low temperatures. The main limitation of this work is twofold: 1) testing on larger base models, and 2) evaluations beyond math and MMLU-STEM, both of which remain for future research.

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

# A APPENDIX

## A.1 RELATED WORK

Recent large language models (LLMs) have been fine-tuned with reinforcement learning (RL) to enhance complex reasoning abilities. OpenAI's *o1* model was among the first to use large-scale RL to encourage chain-of-thought (CoT) reasoning, leading to significant gains on challenging math and coding benchmarks. DeepSeek-R1 demonstrated that pure RL post-training (without supervised warm-up) can directly induce strong reasoning capabilities in LLMs. The model exhibited emergent behaviors like self-verification and multi-step planning, and achieved performance competitive with *o1*. Similarly, the Kimi k1.5 project scaled RL-based training with extremely long context windows and efficient PPO fine-tuning, enabling the model to backtrack and self-correct (Team et al., 2025). These models underscore the growing importance of RL for enhancing reasoning capabilities.

Beyond large proprietary models, several open research efforts have applied RL to improve reasoning. Han et al. (2023) proposed DialCoT-PPO, which transforms problem solving into a dialogue-based reasoning chain and trains the model to optimize these steps. Reinforced Fine-Tuning (ReFT) has been introduced, combining supervised warm-up with PPO-based exploration of diverse reasoning trajectories, significantly improving accuracy on GSM8K, MathQA, and SVAMP Luong et al. (2024). Self-Explore was introduced, where the model identifies and learns from its own mistakes in reasoning paths Hwang et al. (2024). These works demonstrate the versatility of RL in refining reasoning processes across various tasks and model sizes.

While it is commonly assumed that longer responses inherently lead to better reasoning accuracy, empirical findings are mixed. On one hand, increased response length has been associated with improved accuracy (Guo et al., 2025; Yeo et al., 2025; Zeng et al., 2025). It has shown that a collapse in response length can lead to a degradation in performance Yuan et al. (2025). Reward shaping techniques have been employed to encourage longer outputs Ye et al. (2025). On the other hand, several works have found that longer responses do not necessarily correlate with better performance (Zeng et al., 2025; Xie et al., 2025), and reported diminishing returns—and even performance degradation—when responses became excessively long Wu et al. (2025). Our work provides a deeper understanding of the relationship between response length and accuracy, offering new insights into how these two factors are correlated.

Moreover, while long reasoning traces may improve accuracy, they also increase token usage and latency. Recent studies have applied prompting strategies to limit chain of thoughts and analyzed the resulting impact on accuracy (Xu et al., 2025; Renze & Guven, 2024; Jin et al., 2024; Nayab et al., 2024; Muennighoff et al., 2025). It has been shown that each problem has an intrinsic "token complexity": reducing token count below this threshold significantly harms performance. Current prompt-based strategies for conciseness (e.g., "think step by step, but briefly") often fall short of this optimal limit, revealing opportunities for improvement in efficient reasoning Lee et al. (2025).

To improve efficiency, researchers have explored smaller or faster models (e.g., OpenAI's *o1-mini*) and reward shaping during training. A cosine length-scaling reward has been proposed to promote productive CoT reasoning without verbosity Yeo et al. (2025). Others explored long-to-short distillation—training with verbose CoTs for accuracy, then compressing reasoning via model merging or shortest-path sampling. The Kimi team showed that these compressed reasoning models can outperform even GPT-4 on some tasks while using fewer tokens. Our work suggests that a second phase of RL training can substantially shorten response lengths while maintaining accuracy.

## A.2 APPLYING PENALTY AND DISCOUNTING

Although RL training on a small number of sufficiently difficult problems naturally encourages more concise responses, this effect can be further reinforced by introducing an explicit incentive for brevity. In RL, there are two primary methods to promote shorter responses: discounting the return and applying a negative reward per step (or per excessive token use). While both approaches have limitations, the drawbacks of negative rewards are particularly severe. Introducing a step-wise penalty can create strong local optima that hinder effective learning. For instance, the model might prematurely terminate its response to minimize the cumulative penalty rather than fully solving the problem. Moreover, for negative rewards to meaningfully influence learning, they must be large enough to affect decision-making, yet not so dominant that their accumulation overshadows the

positive reward of reaching a correct answer. Given that different problems require varying response lengths, finding an optimal penalty value that works across all cases can be impractical.

Using a discount factor mitigates these issues, as it avoids altering the model's fundamental reasoning process and does not introduce misleading local optima. However, as discussed in Section 4, it inherently introduces an incremental difference between the value of the subsequent steps. This can dilute the return more than expected and effectively shortens the agent's decision horizon, meaning that distant rewards become less influential. If the required chain of thought extends beyond this effective horizon, the agent may fail to recognize the value of a correct answer, leading to potential accuracy degradation. The extent of this adverse effect depends on the chosen discount factor. We therefore recommend to use $\lambda$ to avoid these complications while benefiting the impact, as explained in Section 4.

### A.3 VALUE BEHAVIOUR OF PPO

Due to random initialization, the initial value network produces values near zero. At each training step, the KL penalty encourages alignment with the value network of the previous step. As the value training continues, $V$ gradually moves from zero toward $V_g$. Consequently, the KL penalty drives $V$ toward zero to maintain similarity to the old values, whereas the regression loss pushes it toward $V_g$. This opposition creates an equilibrium point between zero and $V_g$ where the two losses balance each other. If the KL weight is small, this equilibrium point will be close to $V_g$. When $V$ exceeds $V_g$, both losses align, pulling $V$ toward zero and quickly restoring equilibrium. Consequently, when $V_g > 0$, the value will almost surely *underestimate* it, whereas when $V_g < 0$, the value will almost surely *overestimate* it ($V$ will be less negative). The amount of this under/overestimation depend directly on the KL weight.

### A.4 PROBLEMS FROM OLYMPIADBENCH AND AIME USED FOR TRAINING

For the experiment on OlympiadBench, we used problems from the Hugging Face dataset `Hothan/OlympiadBench` with IDs 2231,2237,2240,2245. For the experiments on the AIME24 dataset, we used 4 sets of training examples from the Hugging Face dataset `HuggingFaceH4/aime_2024`. The set with $\bar{p}_a = 0.16$ included problems with IDs 65, 82, 83, 76. The set with $\bar{p}_a = 0.25$ included problems with IDs 65, 64, 61, 76. The set with $\bar{p}_a = 0.375$ included problems with IDs 61, 64, 71, 74. Finally, the fourth set included problems with IDs 71, 74, 82, 86.

## A.5 Advantage Behaviour of a Group in GRPO

Let $x_1, x_2, \ldots, x_N$ be a set of $N$ binary samples where each $x_i \in \{0, 1\}$. Denote by $k$ the number of ones in the set, so the empirical mean is

$$\mu = \frac{1}{N} \sum_{i=1}^{N} x_i = \frac{k}{N}.$$

The population variance is given by

$$\sigma^2 = \frac{1}{N} \sum_{i=1}^{N} (x_i - \mu)^2 = \mu(1 - \mu) = \frac{k}{N}\left(1 - \frac{k}{N}\right),$$

and thus the standard deviation is

$$\sigma = \sqrt{\frac{k}{N}\left(1 - \frac{k}{N}\right)}.$$

We now analyze the range of possible values for $\sigma$.

**Minimum.** The standard deviation is minimized when all samples are identical, i.e., when $k = 0$ or $k = N$. In either case, $\mu \in \{0, 1\}$, and we have

$$\sigma = \sqrt{\mu(1 - \mu)} = 0.$$

**Maximum.** The function $f(p) = p(1 - p)$ achieves its maximum at $p = \frac{1}{2}$, yielding

$$\sigma = \sqrt{\frac{1}{2}\left(1 - \frac{1}{2}\right)} = \frac{1}{2}.$$

This maximum is achieved when $k = \frac{N}{2}$, which is possible only if $N$ is even. For odd $N$, the maximum is attained at $k = \lfloor N/2 \rfloor$ or $k = \lceil N/2 \rceil$, in which case the standard deviation is slightly less than $\frac{1}{2}$:

$$\sigma = \sqrt{\frac{k}{N}\left(1 - \frac{k}{N}\right)}, \quad \text{where } k = \left\lfloor \frac{N}{2} \right\rfloor \text{ or } \left\lceil \frac{N}{2} \right\rceil.$$

Hence, for any set of $N$ binary samples:

$$0 \leq \sigma \leq \frac{1}{2},$$

with the minimum attained when all samples are identical, and the maximum attained when the number of ones and zeros is (nearly) equal.

**Standard Deviation for $k = 1$ and Various $N$.**

$$\sigma = \sqrt{\frac{1}{N}\left(1 - \frac{1}{N}\right)} = \sqrt{\frac{N - 1}{N^2}},$$

we compute the standard deviation for different values of $N$:

- For $N = 8$: $\quad \sigma = \sqrt{\frac{7}{64}} \approx 0.3307$

- For $N = 16$: $\quad \sigma = \sqrt{\frac{15}{256}} \approx 0.2425$

- For $N = 64$: $\quad \sigma = \sqrt{\frac{63}{4096}} \approx 0.1242$

- For $N = 256$: $\quad \sigma = \sqrt{\frac{255}{65536}} \approx 0.0622$

Exactly the same results for $k = N - 1$ and various $N$ since

$$\sigma = \sqrt{\frac{N-1}{N}\left(1 - \frac{N-1}{N}\right)} = \sqrt{\frac{N-1}{N^2}},$$

Using $\sigma$ and $\mu$, we can directly compute $\hat{A}$ in a closed form as follows

- If $r = 1$:
$$\frac{1-\mu}{\sigma} = \frac{1 - \frac{k}{N}}{\sqrt{\frac{k(N-k)}{N^2}}} = \frac{N-k}{\sqrt{k(N-k)}} = \sqrt{\frac{N-k}{k}}.$$

- If $r = 0$:
$$\frac{0-\mu}{\sigma} = \frac{-\frac{k}{N}}{\sqrt{\frac{k(N-k)}{N^2}}} = -\frac{k}{\sqrt{k(N-k)}} = -\sqrt{\frac{k}{N-k}}.$$

Thus,

$$\hat{A} = \frac{r-\mu}{\sigma} = \begin{cases} \sqrt{\dfrac{N-k}{k}}, & \text{if } r = 1 \\[2ex] -\sqrt{\dfrac{k}{N-k}}, & \text{if } r = 0 \end{cases}$$

Table 5 shows $\hat{A}$ for various values of $N$ and $k$.

**Conclusion.** Based on the numbers above, we can conclude that in the case of Dr.GRPO Liu et al. (2025), omitting the division by $\sigma$ **decreases** the advantage, and thus the loss, by nearly a factor of 3 for a group size of 8, and by up to 10 times for group sizes between 64 and 128. However, removing response length from the denominator leads to a substantial **increase** in loss magnitude, ranging from about 1K to 30K, which is far larger than the diminish from removing $\sigma$.

| $N$ | $k$ | $\frac{r-\mu}{\sigma}$ for $r = 1$ | $\frac{r-\mu}{\sigma}$ for $r = 0$ |
|---|---|---|---|
| 8 | 1 | 2.6458 | -0.3780 |
|  | 2 | 1.7321 | -0.5774 |
|  | 3 | 1.2910 | -0.7746 |
|  | 5 | 0.7746 | -1.2910 |
|  | 6 | 0.5774 | -1.7321 |
|  | 7 | 0.3780 | -2.6458 |
| 16 | 1 | 3.8730 | -0.2582 |
|  | 2 | 2.6458 | -0.3780 |
|  | 3 | 2.0801 | -0.4961 |
|  | 13 | 0.4961 | -2.0801 |
|  | 14 | 0.3780 | -2.6458 |
|  | 15 | 0.2582 | -3.8730 |
| 64 | 1 | 7.9373 | -0.1260 |
|  | 2 | 5.5902 | -0.1796 |
|  | 3 | 4.5255 | -0.2182 |
|  | 61 | 0.2182 | -4.5255 |
|  | 62 | 0.1796 | -5.5902 |
|  | 63 | 0.1260 | -7.9373 |
| 256 | 1 | 15.9687 | -0.0626 |
|  | 2 | 11.2900 | -0.0886 |
|  | 3 | 9.2085 | -0.1086 |
|  | 253 | 0.1086 | -9.2085 |
|  | 254 | 0.0886 | -11.2900 |
|  | 255 | 0.0626 | -15.9687 |

Table 5: Values of $\hat{A} = \frac{r-\mu}{\sigma}$ for binary samples, across various values of $N$ and $k$. The sign indicates whether the sample is above or below the mean.

## A.6 Proofs

### A.6.1 Theorem 1 — PPO Loss under GAE

**Part 1.** Recall that

$$\hat{A}_t = \sum_{l=0}^{T-t-1} \lambda^l \delta_{t+l}, \qquad \delta_t = r_t + V(s_{t+1}) - V(s_t), \qquad R := r - V(s_{T-1}),$$

with $r_t = 0$ for $t < T - 1$, $r_{T-1} = r$, $V(s_T) = 0$, and $|\delta_k| \le \epsilon$ for $k < T - 1$. Reindex the double sum:

$$\sum_{t=0}^{T-1} \hat{A}_t = \sum_{t=0}^{T-1} \sum_{l=0}^{T-t-1} \lambda^l \delta_{t+l} = \sum_{k=0}^{T-1} \delta_k \sum_{t=0}^{k} \lambda^{k-t} = \sum_{k=0}^{T-1} \delta_k \sum_{j=0}^{k} \lambda^j \tag{5}$$

The second equality is obtained by setting $k = t + l$ and then swapping the summations ($0 \le t \le k \le T - 1$).

Separate the terminal term $k = T - 1$ to get

$$\sum_{t=0}^{T-1} \hat{A}_t = R \sum_{j=0}^{T-1} \lambda^j + E, \qquad |E| \le \epsilon \sum_{k=0}^{T-2} \sum_{j=0}^{k} \lambda^j.$$

Therefore

$$S = -\frac{1}{T} \left( R \sum_{j=0}^{T-1} \lambda^j + E \right).$$

For $\lambda < 1$, the first part yields $\sum_{j=0}^{T-1} \lambda^j = \frac{1-\lambda^T}{1-\lambda} \to \frac{1}{1-\lambda}$. The second part gives

$$\sum_{k=0}^{T-2} \sum_{j=0}^{k} \lambda^j = \sum_{k=0}^{T-2} \frac{1 - \lambda^{k+1}}{1 - \lambda} = \frac{(T-1) - \sum_{k=1}^{T-1} \lambda^k}{1 - \lambda} < \frac{T-1}{1-\lambda} < \frac{T}{1-\lambda}.$$

It therefore follows

$$S = -\frac{R}{T(1-\lambda)} + O\!\left(\frac{\epsilon}{1-\lambda}\right).$$

For $\lambda = 1$, $\sum_{j=0}^{T-1} 1 = T$ and $\sum_{k=0}^{T-2}(k+1) = \frac{(T-1)T}{2}$, giving

$$S = -R + O\!\left(\frac{\epsilon T}{2}\right).$$

This proves the $S$ part.

In Appendix A.3 we have shown that when $r > 0$, $V$ underestimates and when $r < 0$, $V$ overestimates. Thus, $R$ has the same sign as $r$ and the assumption of this theorem is fair.

**Part 2.** By definition, the per-token loss is defined as

$$L_t = -\alpha_t \hat{A}_t,$$

with

$$\alpha_t = \begin{cases} \min\!\big(\rho_t, \ \mathrm{clip}(\rho_t, 1 - \epsilon_{\mathrm{clip}}, 1 + \epsilon_{\mathrm{clip}})\big), & \hat{A}_t > 0, \\ \max\!\big(\rho_t, \ \mathrm{clip}(\rho_t, 1 - \epsilon_{\mathrm{clip}}, 1 + \epsilon_{\mathrm{clip}})\big), & \hat{A}_t < 0, \end{cases}$$

where $\rho_t = \frac{\pi_\theta(a_t|s_t)}{\pi_{\mathrm{old}}(a_t|s_t)}$. Assume $0 \le \rho_{\min} \le \rho_t \le \rho_{\max} < \infty$. Then,

$$\hat{A}_t > 0 : \ \alpha_t \in [\rho_{\min}, 1 + \epsilon_{\mathrm{clip}}] \ \Rightarrow \ |\alpha_t - 1| \le \max\{1 - \rho_{\min}, \epsilon_{\mathrm{clip}}\};$$
$$\hat{A}_t < 0 : \ \alpha_t \in [1 - \epsilon_{\mathrm{clip}}, \rho_{\max}] \ \Rightarrow \ |\alpha_t - 1| \le \max\{\epsilon_{\mathrm{clip}}, \rho_{\max} - 1\}.$$

Recall that

$$L = -\frac{1}{T} \sum_{t=0}^{T-1} \alpha_t \hat{A}_t, \qquad S = -\frac{1}{T} \sum_{t=0}^{T-1} \hat{A}_t.$$

Therefore, with $\alpha_{\mathrm{dev}} := \max\{1 - \rho_{\min}, \rho_{\max} - 1, \epsilon_{\mathrm{clip}}\}$,

$$|L - S| = \left| -\frac{1}{T} \sum_{t=0}^{T-1} (\alpha_t - 1)\hat{A}_t \right| \le \frac{1}{T} \sum_{t=0}^{T-1} |\alpha_t - 1| \, |\hat{A}_t| \le \alpha_{\mathrm{dev}} \frac{1}{T} \sum_{t=0}^{T-1} |\hat{A}_t|.$$

**Same-sign case.** If all $\hat{A}_t$ share the same sign $\sigma \in \{\pm 1\}$, write $\hat{A}_t = \sigma |\hat{A}_t|$ and set

$$U_S := \frac{1}{T}\sum_{t=0}^{T-1} |\hat{A}_t| \geq 0, \qquad U_L := \frac{1}{T}\sum_{t=0}^{T-1} \alpha_t |\hat{A}_t| \geq 0.$$

Then $S = -\sigma U_S$ and $L = -\sigma U_L$. Therefore, $\operatorname{sgn} L = \operatorname{sgn} S$, assuming $U_S$ and $U_L$ are nonzero.

We get ranges of $\alpha_t$, multiply them by $|\hat{A}_t|$, and then take the average, resulting in the following:

$$\sigma = +1: \quad \rho_{\min} \leq \alpha_t \leq \min(\rho_{\max}, 1 + \epsilon_{\text{clip}}) \;\Rightarrow\; \rho_{\min} U_S \leq U_L \leq \min(\rho_{\max}, 1 + \epsilon_{\text{clip}}) U_S$$

$$\sigma = -1: \quad 1 - \epsilon_{\text{clip}} \leq \alpha_t \leq \rho_{\max} \;\Rightarrow\; (1 - \epsilon_{\text{clip}}) U_S \leq U_L \leq \rho_{\max} U_S$$

Noting that $|S| = U_S \geq 0$ and $|L| = U_L \geq 0$, it implies,

$$\sigma = +1: \quad \rho_{\min} |S| \;\leq\; |L| \;\leq\; \min(\rho_{\max}, 1 + \epsilon_{\text{clip}}) |S|,$$

$$\sigma = -1: \quad (1 - \epsilon_{\text{clip}}) |S| \;\leq\; |L| \;\leq\; \rho_{\max} |S|.$$

If $\rho_{\min} = 0$, the lower bound in the $\sigma = +1$ line becomes the trivial $0 \leq |L|$.

This completes the proof of Theorem 1.

The next result provides a sufficient condition for $\hat{A}_t$ to have the same sign. Note that this condition is not necessary and is somehow strong.

### A.7 PER-TOKEN FIXED-SIGN CONDITION

**Lemma 1 [Per-token fixed-sign condition]:** Let $m := T - t$ and $t \in \{1, \ldots, T\}$. Then

$$\hat{A}_t \;=\; R\lambda^{m-1} + E_t, \qquad |E_t| \;\leq\; \epsilon \sum_{j=0}^{m-2} \lambda^j = \begin{cases} \epsilon \dfrac{1 - \lambda^{m-1}}{1 - \lambda}, & \lambda < 1, \\ \epsilon(m-1), & \lambda = 1. \end{cases}$$

Consequently,

$$\operatorname{sgn} \hat{A}_t = \operatorname{sgn} R \quad \text{whenever} \quad |R|\lambda^{m-1} \;>\; \epsilon \sum_{j=0}^{m-2} \lambda^j.$$

A uniform (in $t$) sufficient condition is

$$|R| \;>\; \epsilon \cdot \Phi_{\lambda, T}, \qquad \Phi_{\lambda, T} := \begin{cases} \dfrac{1 - \lambda^{T-1}}{(1 - \lambda)\lambda^{T-1}}, & \lambda < 1, \\ T - 1, & \lambda = 1. \end{cases}$$

**Proof.** Split the GAE sum at the terminal step:

$$\hat{A}_t = \sum_{l=0}^{m-2} \lambda^l \delta_{t+l} + \lambda^{m-1} \delta_{T-1} = E_t + \lambda^{m-1} R.$$

If $|\delta_k| \leq \epsilon$ for all $k < T - 1$, then $|E_t| \leq \epsilon \sum_{l=0}^{m-2} \lambda^l$; the closed forms follow from the geometric sum when $\lambda < 1$, and from counting terms when $\lambda = 1$. If $|R|\lambda^{m-1} > |E_t|$, then $R\lambda^{m-1}$ dominates $E_t$, hence $\operatorname{sgn} \hat{A}_t = \operatorname{sgn} R$. For a uniform condition over $t$, note that the right-hand side increases with $m$, so the worst case is $m = T$, yielding the stated $\Phi_{\lambda, T}$.

#### A.7.1 THEOREM 2 — GRPO PROLIXITY

Since $\hat{A}_t^{(i)} = \hat{A}^{(i)}$, we rewrite equation 2 as

$L_{\text{GRPO}}(\pi_\theta) =$

$$\begin{cases} -\mathbb{E}_{q \sim p_Q, \{o^{(i)}\}_{i=0}^{G-1} \sim \pi_{\theta_{\text{old}}}} \left[ \dfrac{1}{G}\sum_{i=0}^{G-1} \dfrac{\hat{A}_i}{|o^{(i)}|} \sum_{t=0}^{|o^{(i)}|-1} \min\left( \dfrac{\pi_\theta(o_t^{(i)}|q, o_{<t}^{(i)})}{\pi_{\theta_{\text{old}}}(o_t^{(i)}|q, o_{<t}^{(i)})}, 1 + \epsilon \right) \right], & \text{if } \hat{A}_i > 0 \\[4mm] -\mathbb{E}_{q \sim p_Q, \{o^{(i)}\}_{i=0}^{G-1} \sim \pi_{\theta_{\text{old}}}} \left[ \dfrac{1}{G}\sum_{i=0}^{G-1} \dfrac{\hat{A}_i}{|o^{(i)}|} \sum_{t=0}^{|o^{(i)}|-1} \max\left( \dfrac{\pi_\theta(o_t^{(i)}|q, o_{<t}^{(i)})}{\pi_{\theta_{\text{old}}}(o_t^{(i)}|q, o_{<t}^{(i)})}, 1 - \epsilon \right) \right], & \text{if } \hat{A}_i < 0 \end{cases}$$

Without loss of generality, consider a fixed group and focus on a single response with length $T$ within the group. We examine how the loss behaves under two scenarios: in the first, the policy assigns a high probability of terminating the response at token $T-1$; in the second, it assigns a low probability of terminating at $T-1$ but instead continues with one additional token, terminating at token $T$, with a probability comparable to the first scenario's stopping probability at $T-1$. The goal is to understand how the loss changes under these two behaviors, depending on whether the advantage is positive or negative.

Since the group is fixed, we only need to consider the contribution of this sample in the loss; we call it $L$. To simplify the notation, for the chosen response, define $o_t$ as the $t$-th token. The theorem only concern token $T-1$ and how it impacts the loss. Hence, by construction, all tokens up to and including $T-2$ are **fixed** and the expectation only affects token $T-1$. With notation simplifications, it therefore deduces

$$
L(\pi_\theta) = \begin{cases} -\frac{\hat{A}}{T}\mathbb{E}_{o_{T-1}\sim\pi_{\theta_{\text{old}}}}\left[\sum_{t=0}^{T-1}\min\left(\frac{\pi_\theta(o_t)}{\pi_{\theta_{\text{old}}}(o_t)}, 1+\epsilon\right)\right], & \text{if } \hat{A} > 0 \\[2mm] -\frac{\hat{A}}{T}\mathbb{E}_{o_{T-1}\sim\pi_{\theta_{\text{old}}}}\left[\sum_{t=0}^{T-1}\max\left(\frac{\pi_\theta(o_t)}{\pi_{\theta_{\text{old}}}(o_t)}, 1-\epsilon\right)\right], & \text{if } \hat{A} < 0 \end{cases}
$$

where $\hat{A}$ is the advantage of the selected response. Separating the first $T-1$ fixed tokens, it yields

$$
L(\pi_\theta) = \begin{cases} -\frac{\hat{A}}{T}\sum_{t=0}^{T-2}\min\left(\frac{\pi_\theta(o_t)}{\pi_{\theta_{\text{old}}}(o_t)}, 1+\epsilon\right) - \frac{\hat{A}}{T}\mathbb{E}_{o_{T-1}\sim\pi_{\theta_{\text{old}}}}\min\left(\frac{\pi_\theta(o_{T-1})}{\pi_{\theta_{\text{old}}}(o_{T-1})}, 1+\epsilon\right), & \text{if } \hat{A} > 0 \\[2mm] -\frac{\hat{A}}{T}\sum_{t=0}^{T-2}\max\left(\frac{\pi_\theta(o_t)}{\pi_{\theta_{\text{old}}}(o_t)}, 1-\epsilon\right) - \frac{\hat{A}}{T}\mathbb{E}_{o_{T-1}\sim\pi_{\theta_{\text{old}}}}\max\left(\frac{\pi_\theta(o_{T-1})}{\pi_{\theta_{\text{old}}}(o_{T-1})}, 1-\epsilon\right), & \text{if } \hat{A} < 0 \end{cases}
$$

$$
= \begin{cases} -\frac{\hat{A}}{T}\beta_1 - \frac{\hat{A}}{T}\mathbb{E}_{o_{T-1}\sim\pi_{\theta_{\text{old}}}}\min\left(\frac{\pi_\theta(o_{T-1})}{\pi_{\theta_{\text{old}}}(o_{T-1})}, 1+\epsilon\right), & \text{if } \hat{A} > 0 \\[2mm] -\frac{\hat{A}}{T}\beta_2 - \frac{\hat{A}}{T}\mathbb{E}_{o_{T-1}\sim\pi_{\theta_{\text{old}}}}\max\left(\frac{\pi_\theta(o_{T-1})}{\pi_{\theta_{\text{old}}}(o_{T-1})}, 1-\epsilon\right), & \text{if } \hat{A} < 0 \end{cases} \tag{6}
$$

where $\beta_1 > 0$ and $\beta_2 > 0$ are the first terms of the two equations, respectively. By assumption, $\pi_\theta$ and $\pi_{\theta_{\text{old}}}$ are identical for $t \leq T-2$. Hence, $\beta_1 = \beta_2 = B = T-1$. Note that in general $0 < \beta_1 \leq (T-1)(1+\epsilon)$ and $\beta_2 \geq (T-1)(1-\epsilon)$. However, even in the general case, the KL regularization should naturally keep most sampling ratios close to one, meaning that both $\beta$s should remain close to $T-1$.

Recall that the response is sampled from $\pi_{\theta_{\text{old}}}$. Let $\tau$ denote the terminal token. Assume that $\pi_{\theta_{\text{old}}}(o_{T-1} = \tau) \neq 0$, and that this probability is neither too small nor too large. Let a trajectory sampled from $\pi_{\theta_{\text{old}}}$ produce a sequence of length $T$, meaning $o_{T-1} = \tau$. We consider a new policy $\pi_\theta$ that matches $\pi_{\theta_{\text{old}}}$ exactly for all $t \leq T-2$, differing only at the final token. Our objective is to analyze how modifying $\pi_\theta$ in this way affects the loss, under the conditions $\hat{A} > 0$ and $\hat{A} < 0$.

**Case 1 ($\hat{A} > 0$):** If $\pi_\theta(o_{T-1} = \tau)$ decreases relative to $\pi_{\theta_{\text{old}}}(o_{T-1} = \tau)$, then the ratio $\frac{\pi_\theta}{\pi_{\theta_{\text{old}}}}$ becomes smaller, which is favored by the $\min$ operator in Equation equation 6. Consequently, the loss increases (becomes less negative). Conversely, if $\pi_\theta(o_{T-1} = \tau)$ increases, the loss decreases (up to $1+\epsilon$). Therefore, when $\hat{A} > 0$, GRPO encourages raising $\pi_\theta(o_{T-1} = \tau)$, effectively increasing the probability of termination and discouraging longer sequences.

**Case 2 ($\hat{A} < 0$):** If $\pi_\theta(o_{T-1} = \tau)$ increases relative to $\pi_{\theta_{\text{old}}}(o_{T-1} = \tau)$, then the ratio $\frac{\pi_\theta}{\pi_{\theta_{\text{old}}}}$ can grow unboundedly, which is favored by the $\max$ operator in Equation equation 6. Since $-\hat{A} > 0$, this results in an increase in the loss. Conversely, if $\pi_\theta(o_{T-1} = \tau)$ decreases, the loss is reduced, down to $1-\epsilon$. Therefore, when $\hat{A} < 0$, GRPO encourages reducing $\pi_\theta(o_{T-1} = \tau)$, effectively lowering the probability of termination and promoting longer sequences.

### A.7.2 THEOREM 3 — GENERAL CONCISENESS

Let $\pi_\theta$ be a parameterized stochastic policy, and $\pi_{\theta_{\text{old}}}$ be the policy before the update. If the advantage estimate for response $i$ is positive, i.e., $\hat{A}^{(i)} > 0$, both PPO and GRPO optimize a clipped surrogate

objective:

$$L^{(i)}(\theta) = -\hat{A}^{(i)} \cdot \frac{1}{T_i} \sum_{t=0}^{T_i-1} \min\left(\rho_t^{(i)}, 1+\epsilon\right), \quad \rho_t^{(i)} = \frac{\pi_\theta(o_t^{(i)} \mid o_{<t}^{(i)})}{\pi_{\theta_{\text{old}}}(o_t^{(i)} \mid o_{<t}^{(i)})},$$

where $T_i$ is the length of response $i$, and $\epsilon > 0$ is a clipping threshold. This loss term then contributes to the total average loss, which is minimized by the optimizer.

The loss is a function of $\pi_\theta$, but the trajectories are sampled from $\pi_{\text{old}}$, which is fixed during optimization. Therefore, the gradient is computed with respect to the numerator of $\rho_t^{(i)}$, treating $\pi_{\text{old}}$ as a constant (for clarity, we drop all the conditioning in the policies):

$$\nabla_\theta \rho_t^{(i)} = \nabla_\theta \left(\frac{\pi_\theta(o_t^{(i)})}{\pi_{\theta_{\text{old}}}(o_t^{(i)})}\right) = \frac{1}{\pi_{\theta_{\text{old}}}(o_t^{(i)})} \nabla_\theta \pi_\theta(o_t^{(i)}) = \rho_t^{(i)} \nabla_\theta \log \pi_\theta(o_t^{(i)}) \tag{7}$$

We break the loss into per-tokens, $L^{(i)} = \sum_{t=0}^{T-1} L_t^{(i)}$. Therefore, for token $t$ we write

$$\nabla_\theta L_t^{(i)} = -\hat{A}^{(i)} \cdot \frac{1}{T_i} \nabla_\theta \min(\rho_t^{(i)}, 1+\epsilon).$$

Importantly, this gradient characterizes how token $t$ is selected after the update. If $\rho_t < 1+\epsilon$, the min operator vanishes and equation equation 7 yields

$$\nabla_\theta L_t^{(i)} = -\hat{A}^{(i)} \cdot \frac{1}{T} \nabla_\theta \rho_t^{(i)} = -\hat{A}^{(i)} \cdot \frac{1}{T} \cdot \rho_t^{(i)} \nabla_\theta \log \pi_\theta(o_t^{(i)}) \tag{8}$$

Let us focus on a fixed position $t$ of the sequence. Assume that the last layer of $\boldsymbol{\pi_\theta}$ is a softmax over $K$ tokens with logits $\boldsymbol{z(\theta)} \in \mathbb{R}^K$. The probability assigned to token $k$ is:

$$\pi_\theta(o_t^{(i)} = k) = \frac{e^{z_k}}{\sum_{j=1}^K e^{z_j}}.$$

where $\pi_\theta(k)$ is a short-hand for $\pi_\theta(o_t^k \mid o_{<t}^k)$ for a fixed $t$. To compute the gradient of the log-probability, we write

$$\log \pi_\theta(o_t^{(i)} = k) = z_k - \log\left(\sum_{j=1}^K e^{z_j}\right),$$

Differentiating with respect to $z$ yields

$$\nabla_z \log \pi_\theta(o_t^{(i)} = k) = \nabla_z z_k - \nabla_z \log\left(\sum_{j=1}^K e^{z_j}\right).$$

The first term is simply $\boldsymbol{e_k}$, the one-hot vector with 1 in position $k$. The second term is the gradient of the log-partition function, which equals the softmax vector,

$$\nabla_z \log\left(\sum_{j=1}^K e^{z_j}\right) = \left(\frac{e^{z_1}}{\sum_j e^{z_j}}, \ldots, \frac{e^{z_K}}{\sum_j e^{z_j}}\right) = \boldsymbol{\pi_\theta}.$$

So we get

$$\nabla_z \log \pi_\theta(k) = \boldsymbol{e_k} - \boldsymbol{\pi_\theta}.$$

This gradient is a vector in $\mathbb{R}^K$, and its squared norm is

$$\|\nabla_z \log \pi_\theta(o_t^{(i)} = k)\|^2 = \sum_{j=1}^K \left(\delta_{jk} - \pi_\theta(o_t^{(i)} = j)\right)^2 = \left(1 - \pi_\theta(o_t^{(i)} = k)\right)^2 + \sum_{j \neq k} \pi_\theta(o_t^{(i)} = j)^2 \tag{9}$$

with $\delta_{jk}$ denoting the Kronecker delta function. Defining $f(k) := \|\nabla_z \log \pi_\theta(k)\|$, equation equation 9 proves part 1 of the theorem.

For the next part, let us focus on the loss of the **first token**. By assumption, clipping is inactive for the first tokens of both sequences (i.e., $\rho_0^{(S)}, \rho_0^{(L)} < 1 + \epsilon$). Thus, from equation 8, the gradient of the surrogate loss for token $o_0^{(i)}$ with respect to model parameters $\theta$ is:

$$\nabla_\theta L^{(i)}(o_0^{(i)}) = -\frac{\hat{A}}{T_i} \rho_0^{(i)} \nabla_\theta \log \pi_\theta(o_0^{(i)}).$$

Since the policy $\pi_\theta$ is computed via a softmax over logits $z(\theta) \in \mathbb{R}^K$, the chain rule yields:

$$\nabla_\theta \log \pi_\theta(o_0^{(i)}) = \left(\frac{\partial z}{\partial \theta}\right)^\top \nabla_z \log \pi_\theta(o_0^{(i)}).$$

Thus, the full gradient norm becomes

$$\left\|\nabla_\theta L^{(i)}(o_0^{(i)})\right\| = \frac{\hat{A}}{T_i} \rho_0^{(i)} \left\|\left(\frac{\partial \mathbf{z}}{\partial \boldsymbol{\theta}}\right)^\top \nabla_z \log \pi_\theta(o_0^{(i)})\right\| \tag{10}$$

Now consider comparing $\left\|\nabla_\theta L^{(S)}(o_0^{(S)})\right\|$ and $\left\|\nabla_\theta L^{(L)}(o_0^{(L)})\right\|$. These gradients differ in two ways: the softmax log-prob gradients, and the Jacobians of the logits w.r.t. $\theta$.

However, at the first token position ($t = 0$), both sequences share the same context (i.e., the empty prefix or prompt), so the network's hidden activations leading up to $z$ are identical. The only variation is the sampled token at $t = 0$, which does not affect the computation of $z_0$. Therefore, we have:

$$\left(\frac{\partial \mathbf{z}}{\partial \boldsymbol{\theta}}\right)^{(S)} = \left(\frac{\partial \mathbf{z}}{\partial \boldsymbol{\theta}}\right)^{(L)} := \mathbf{J}$$

where we define $\mathbf{J} \in \mathbb{R}^{K \times N}$ to be the common Jacobian matrix with $K$ representing the vocabulary size, as before, and $N$ the dimensionality of the parameter space, with the corresponding parameter vector $\boldsymbol{\theta} \in \mathbb{R}^N$.

From expression equation 10, this yields

$$\left\|\nabla_\theta L^{(S)}(o_0^{(S)})\right\| > \left\|\nabla_\theta L^{(L)}(o_0^{(L)})\right\|$$

$$\iff \quad \frac{\hat{A}}{T_S} \rho_0^{(S)} \left\|\mathbf{J}^\top \nabla_z \log \pi_\theta(o_0^{(S)})\right\| > \frac{\hat{A}}{T_L} \rho_0^{(L)} \left\|\mathbf{J}^\top \nabla_z \log \pi_\theta(o_0^{(L)})\right\|$$

$\hat{A} > 0$ cancels out without changing the inequality's direction. Using the previous results, it deduces

$$\frac{\rho_0^{(S)}}{T_S} \left\|\mathbf{J}^\top(\mathbf{e}_{k_S} - \boldsymbol{\pi}_\theta)\right\| > \frac{\rho_0^{(L)}}{T_L} \left\|\mathbf{J}^\top(\mathbf{e}_{k_L} - \boldsymbol{\pi}_\theta)\right\| \tag{11}$$

From the singular value bounds for $\mathbf{J}$, we have

$$\sigma_{\min}(\mathbf{J}) \|\mathbf{v}\| \le \left\|\mathbf{J}^\top \mathbf{v}\right\| \le \sigma_{\max}(\mathbf{J}) \|\mathbf{v}\|,$$

where

- $\sigma_{\min}(\mathbf{J})$ is the smallest singular value,
- $\sigma_{\max}(\mathbf{J})$ is the largest singular value,
- $\kappa(\mathbf{J}) = \frac{\sigma_{\max}(\mathbf{J})}{\sigma_{\min}(\mathbf{J})}$ is the condition number of $\mathbf{J}$.

Using these bounds, inequality equation 11 can be written as

$$\frac{\rho_0^{(S)}}{T_S} \|\mathbf{e}_{k_S} - \boldsymbol{\pi}_\theta\| > \frac{\rho_0^{(L)}}{T_L} \tilde{\kappa} \|\mathbf{e}_{k_L} - \boldsymbol{\pi}_\theta\|.$$

where $\tilde{\kappa} \in [\kappa(\mathbf{J})^{-1}, \ \kappa(\mathbf{J})]$ is a constant. Substituting $f(k) := \|\mathbf{e}_k - \boldsymbol{\pi}_\theta\|$ from the previous part yields the claimed inequality and completes the second part of the proof.

**Sufficient condition:**

$$\frac{\rho_0^{(S)}}{T_S}\, \sigma_{\min}(\mathbf{J})\, \|\mathbf{e}_{k_S} - \boldsymbol{\pi}_\theta\| > \frac{\rho_0^{(L)}}{T_L}\, \sigma_{\max}(\mathbf{J})\, \|\mathbf{e}_{k_L} - \boldsymbol{\pi}_\theta\|$$

or equivalently,

$$\frac{\rho_0^{(S)}}{T_S}\, \|\mathbf{e}_{k_S} - \boldsymbol{\pi}_\theta\| > \frac{\rho_0^{(L)}}{T_L}\, \kappa(\mathbf{J})\, \|\mathbf{e}_{k_L} - \boldsymbol{\pi}_\theta\|\,.$$

**Necessary condition:**

$$\frac{\rho_0^{(S)}}{T_S}\, \sigma_{\max}(\mathbf{J})\, \|\mathbf{e}_{k_S} - \boldsymbol{\pi}_\theta\| > \frac{\rho_0^{(L)}}{T_L}\, \sigma_{\min}(\mathbf{J})\, \|\mathbf{e}_{k_L} - \boldsymbol{\pi}_\theta\|$$

or equivalently,

$$\frac{\rho_0^{(S)}}{T_S}\, \|\mathbf{e}_{k_S} - \boldsymbol{\pi}_\theta\| > \frac{\rho_0^{(L)}}{T_L}\, \kappa(\mathbf{J})^{-1}\, \|\mathbf{e}_{k_L} - \boldsymbol{\pi}_\theta\|\,.$$

Remark that:

1. The necessary condition is almost trivial, since $\kappa(\mathbf{J})^{-1}$ can be very small, helping equation 11 to easily hold.

2. The sufficient condition is too strong and may not be needed.

3. In practice, equation 11 can hold with $\tilde{\kappa}$ significantly smaller than $\kappa(\mathbf{J})$.

**Effect of Temperature.** If the logits are scaled by temperature $\tau > 0$ before softmax:

$$\pi_\theta(k) = \frac{e^{z_k/\tau}}{\sum_{j=1}^{K} e^{z_j/\tau}},$$

then the gradient becomes:

$$\nabla_z \log \pi_\theta(k) = \frac{1}{\tau}(\boldsymbol{e_k} - \boldsymbol{\pi_\theta}),$$

and the norm scales as $1/\tau$. However, the form of the inequality and comparisons remain valid modulo this shared scaling, which concludes the last part of the Theorem.

## A.8 EXPERIMENTAL SETUP

For the PPO experiments, we used the following reward scheme: +1 if the final answer was correct and enclosed in a box; –0.5 if the answer was boxed but incorrect; and –1 if no boxed answer was provided. During training, we set $\gamma = 1$ and $\lambda = 0.95$. For each input, the model generated 8 samples using a temperature of 0.6 and top_p $= 1$, with a maximum sequence length of 20,000 tokens. We used the Adam optimizer with a learning rate of $5 \times 10^{-7}$ for the actor and $9 \times 10^{-6}$ for the critic.

For GRPO experiments, we used a +1 reward for a correct and boxed final answer, and a 0 reward otherwise. During training, we set $\gamma = 1$. For each input, the model generated 8 samples using a temperature of 0.6 and top_p $= 1$, with a maximum sequence length of 24,576 tokens. We used the Adam optimizer with a learning rate of $1 \times 10^{-6}$ for the actor.

## A.9 COMPUTE RESOURCES

The experiments were conducted using H100 GPUs for compute. The training was done using 8 GPUs, with each GPU having 80 GB of memory, and sufficient storage to accommodate the dataset and model checkpoints. The second-stage post-training of R1 models required fewer than 60 training steps and took approximately 19 GPU hours for the 1.5B model and 41 GPU hours for the 7B model.

## A.10 STABLE TRAINING OF PPO WITH $\lambda < 1$

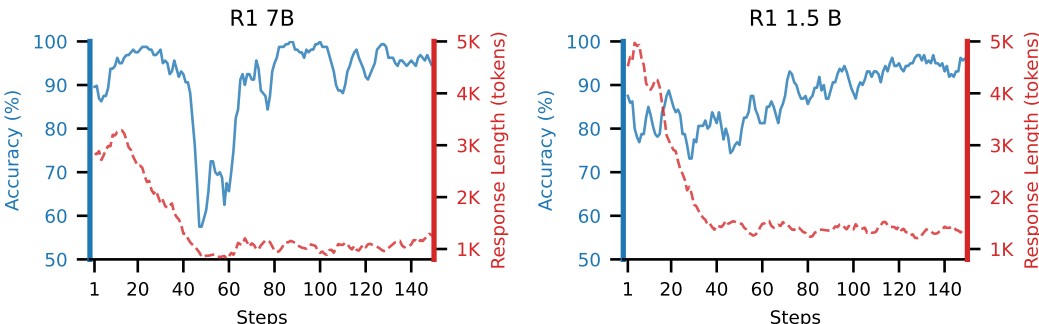

Figure 5: Reinforcement learning is performed using PPO algorithm with 8 example and $\gamma = 1$ on two base models: *DeepSeek-R1-Distill-Qwen-1.5B* (right) and *DeepSeek-R1-Distill-Qwen-7B* (left). The examples are randomly selected from level-5 questions of the MATH dataset. The plots illustrate average accuracy and length of generated responses during the PPO training.

## A.11  UNSTABLE TRAINING OF PPO WITH $\lambda = 1$

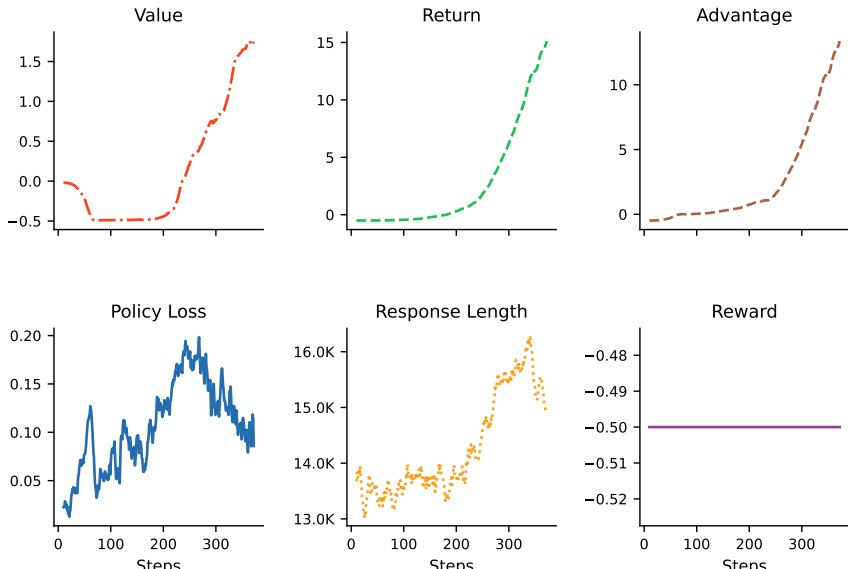

Figure 6: PPO training was conducted with $\lambda = 1$ on four problems selected from the OlympiadBench dataset. Notably, there is an exponential **return overflow** starting around step 100. Importantly, the reward consistently *remains at $-0.5$ over all the training steps*.

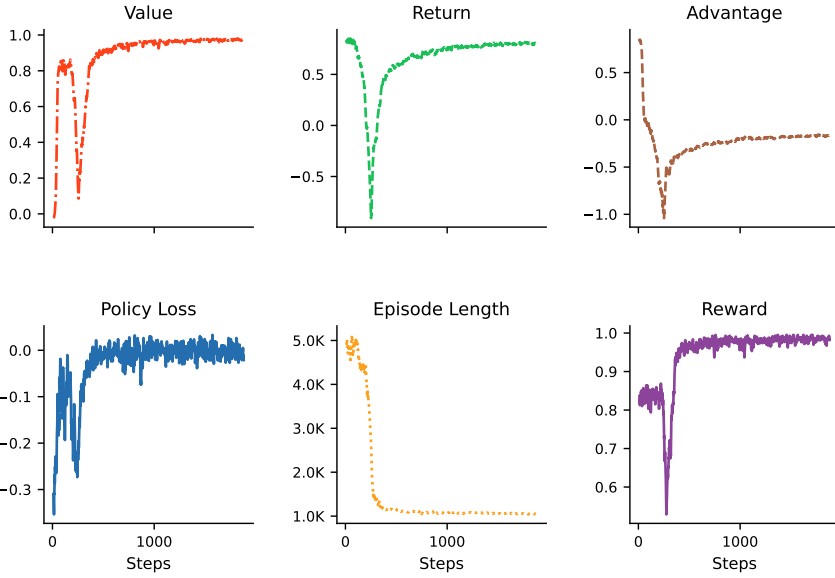

Figure 7: PPO training with $\lambda = 1$, conducted on four problems selected from the MATH dataset, which are somehow solvable. **Return underflow** starts early, but almost quickly stops. Nonetheless, while the response length decreases, it still requires **significantly more steps** to achieve this compared to similar training with $\lambda < 1$ (e.g., compare it to Fig. 5 where in only $\sim 40$ steps, the model achieves similar length reduction).

## A.12 GRPO TRAINING ON MATH EXAMPLES

Fig. 8 illustrates the accuracy and response length of DeepSeek-R1-Distill-Qwen-1.5B over training steps trained with GRPO using eight training examples from the training subset of MATH dataset and, evaluated on different *test* datasets, AIME 2024, AMC 2023, and MATH-500. For evaluation, we generated four samples per query using a temperature of $0.6$ and top-p of $0.95$. In the first half of the steps, response length decreased while accuracy remained stable or improved across benchmarks. In the second half, both response length and accuracy showed fluctuations, indicating unstable performance.

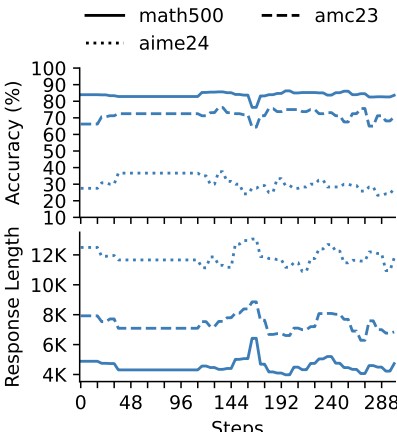

Figure 8: Response dynamics of *DeepSeek-R1-Distill-Qwen-1.5B* trained with GRPO using 8 problems from the level-5 subset of MATH dataset evaluated on three mathematics benchmarks (different line styles).

