# OpenReview forum: "Concise Reasoning via Reinforcement Learning"
_ICLR.cc/2026/Conference — ICLR 2026 Conference Withdrawn Submission_

### Official Review · Reviewer_qJt6 · 2025-10-30

**Soundness:** 3
**Presentation:** 3
**Contribution:** 2
**Rating:** 4
**Confidence:** 3

**Summary:**

This paper tackles the problem of verbosity and excessive token usage in LLMs, reframing response lengthening as an optimization artifact rather than a sign of deeper reasoning. The authors provide a theoretical analysis showing that PPO and GRPO tend to increase response length when faced with negative rewards. Building on this insight, they propose a simple two-phase reinforcement learning strategy: (1) train reasoning capability on general problems, and (2) fine-tune conciseness using a small set of solvable problems. The method achieves substantial length reduction—over 50% for a 1.5B model—while maintaining or improving accuracy.

**Strengths:**

1. The paper verbosity arises from reinforcement learning dynamics rather than reasoning necessity, which is both novel and well-substantiated. The theoretical analysis clearly shows how PPO and GRPO loss functions inherently favor longer responses under failure, providing an elegant and insightful explanation for a long-observed phenomenon.

2. The proposed two-phase RL strategy directly follows from the theory and is both intuitive and easy to implement. Separating reasoning acquisition and conciseness tuning leads to a clean training recipe. Using a small set of solvable problems to reverse verbosity is a clever and practical idea.

3. Experiments demonstrate a rare win-win in efficiency and accuracy. The second phase reduces response length by over 54% for the 1.5B model and 40% for the 7B model while preserving or improving accuracy on multiple math benchmarks. The robustness to low-temperature decoding further reinforces the model’s stability.

4. The paper highlights that training on as few as four solvable problems can yield major accuracy gains (up to 30%) for models without prior RL tuning. The approach generalizes across tasks, suggesting that the learned conciseness principle is not domain-specific.

**Weaknesses:**

1. The conciseness phase relies on an extremely small, manually curated dataset (e.g., eight problems from MATH). It is unclear how the model generalizes a broad conciseness skill from such limited data rather than memorizing stylistic features.

2. Selecting “occasionally solvable” problems is critical for success, yet the procedure is only briefly described and not automated. This limits reproducibility and scalability to new domains.

3. The first phase is assumed to correspond to off-the-shelf RL models like DeepSeek-R1. Without running Phase 1 in-house, the study cannot directly quantify how much verbosity arises during initial training or how much Phase 2 mitigates it.

4. Although the paper theoretically explains why GRPO fails to enforce conciseness, it does not empirically confirm this failure. A direct comparison would strengthen the argument and validate the theoretical claims.

**Questions:**

1. How does the model generalize conciseness from such a small Phase 2 dataset without overfitting to those few examples?

2. What criteria were used to identify the “occasionally solvable” problems, and how sensitive is performance to their selection?

---

### Official Review · Reviewer_oYmF · 2025-10-31

**Soundness:** 3
**Presentation:** 2
**Contribution:** 3
**Rating:** 4
**Confidence:** 2

**Summary:**

The paper presents an analysis of modern reinforcement learning methods for language models, specifically PPO and GRPO. Following a theoretical analysis, the authors demonstrate that both GRPO and PPO tend to produce long responses on tasks that the model cannot solve. To address this, they propose a two-stage training scheme: in the second stage, the model is fine-tuned on a small set of examples that it can already solve correctly. This approach reduces the length of reasoning traces by approximately half while maintaining performance.

**Strengths:**

- Although the difference in response length between correct and incorrect solutions is a known phenomenon, the paper’s theoretical analysis is valuable and may inform the development of new methods.
- The proposed two-stage scheme effectively reduces reasoning trace length by roughly a factor of two.
- The paper includes a comprehensive analysis across samples of varying complexity.

**Weaknesses:**

- The observed dynamics are demonstrated only for Qwen-2.5–based models. However, this model family is known to exhibit atypical behavior during RL fine-tuning (see [1]). Additional experiments on other architectures are needed to assess generalizability.
- Results are limited to mathematical reasoning benchmarks. It remains unclear whether the same approach would generalize to other domains, such as code generation, or whether different types of examples would be required.

Minor:

- The first claim in the Introduction suggests that this is the first work to observe the difference in length between correct and incorrect responses. However, this phenomenon has been noted previously (e.g., in [2]). The authors should either reword this statement or provide justification for their claim.
- The symbols $ \lambda $ and $ \gamma $ are referenced before being formally introduced. Although these are standard hyperparameters in GAE, it would be helpful to define them explicitly by including the advantage estimation equation before.

[1] Spurious Rewards: Rethinking Training Signals in RLVR. Rulin Shao, Shuyue Stella Li et al.

[2] SEAL: Steerable Reasoning Calibration of Large Language Models for Free. Runjin Chen, Zhenyu Zhang et al.

**Questions:**

See weaknesses

---

### Official Review · Reviewer_Qk2x · 2025-10-31

**Soundness:** 3
**Presentation:** 2
**Contribution:** 2
**Rating:** 2
**Confidence:** 3

**Summary:**

The paper investigates methods for reducing the length of reasoning traces in language models through reinforcement learning. It presents theoretical analysis of aspects of PPO and GRPO, and an experimental evaluation showing that it is possible to reduce length by training on only 8 examples in the tested setting.

**Strengths:**

- The paper deals with a timely and interesting topic.
- The observation that it is possible to get length reductions by training on only 8 examples is interesting.

**Weaknesses:**

- The paper focuses on too many things at once. The paper tries to combine empirical observations about length-accuracy correlation (section 2), theoretical analysis of PPO and GRPO (section 4, section 5), and a demonstration that length reductions are possible using very small amounts of training examples, but these components don't form a cohesive narrative.
- I am not sure about the purpose of the analysis in section 2. It shows that reasoning traces for correct problems tend to be shorter than those for incorrect problems, which is not surprising. Among correct problems, it is still possible that more difficult problems tend to have longer reasoning traces (indeed we see this in Table 1, AIME Correct are longer than MATH 500 Correct), suggesting that there could still be some relationship between difficulty and the expected length of a reasoning trace.
- The paper presents theoretical results in section 4 about PPO, but it is unclear what the key insight is. There are a list of bullet points, but none seem particularly insightful.
- Theorem 2 does not seem surprising: if the advantage of an action is negative, we would expect that decreasing the probability of the action would reduce loss (and analogously for positive advantages).
- It's unclear why the experiments are done on 8 examples. While this does seem interesting, it's disconnected from the previous sections of the paper. It's also not clear how the experiments generalize to other data distributions or other base models.
- The discussion of related work is relegated to the appendix. Various statements in the paper have missing citations.

**Questions:**

Please respond to the above, including:
- Can you clarify the main message of the paper? Consider focusing the paper on a more cohesive narrative that better connects the theoretical analysis with the experimental setup.
- Can you clarify the contribution and purpose of section 2? The claimed conclusion about length-accuracy correlation not being an artifact of problem difficulty does not follow from the presented analysis.
- What is the key insight from the theoretical results in section 4? The current presentation makes it difficult to understand the main takeaway.
- How do the experiments conducted on only 8 examples fit with the rest of the paper? Do the results to generalize to different data distributions and base models?

---

### Official Review · Reviewer_YWER · 2025-11-01

**Soundness:** 2
**Presentation:** 3
**Contribution:** 3
**Rating:** 4
**Confidence:** 4

**Summary:**

The work offers a clear and intuitive theoretical account connecting PPO/GAE dynamics to response length, formalizing an empirical phenomenon many practitioners observe. The MDP framing for single problems and the asymptotic characterization of the averaged PPO term with λ<1 are insightful and yield testable predictions about length dynamics under positive vs. negative returns. The analysis of GRPO’s group-normalized advantage clarifies collapse modes and explains why conciseness can stall in all-correct or all-incorrect groups. The proposed two-phase RL procedure is simple, inexpensive, and practically actionable, and the paper reports consistent length reductions without sacrificing accuracy, along with improved low-temperature robustness. The writing is generally clear, with derivations that are easy to follow and practical recommendations that could influence training practice.

**Strengths:**

1. **Theoretical clarity and practical intuition**: The PPO/GAE asymmetry neatly links return sign to length dynamics and explains widely observed behavior. The λ=1 noise amplification discussion is useful for practitioners.
2. **GRPO insight**: The analysis of group‑normalized advantage explains why training can enter collapse regimes (all‑correct/all‑incorrect batches), and why conciseness may not be reliably induced by GRPO alone.
3. **Simple, low‑cost recipe**: The two‑phase method is easy to adopt; empirical results show consistent length reductions with negligible or positive accuracy changes and substantially better τ=0 robustness.
4. **Actionable guidance**: Prefer PPO with λ<1; avoid per‑step length penalties; use a small “occasionally solvable” Phase‑2 set; monitor policy loss–length coupling; watch for GRPO advantage collapse dominated by KL.

**Weaknesses:**

1. **Limited closed‑loop evidence for the two‑phase paradigm**: While the paper includes a PPO run on unsolvable items showing length increase and a difficulty‑bucket study (7.1) consistent with theory, it lacks a comprehensive Phase‑1→Phase‑2 “length up then down” demonstration on larger sets with multiple seeds and uncertainty estimates.
2. **Small data and missing scaling curves**: Phase‑2 often uses 4–8 problems. There are no curves vs. Phase‑2 size (e.g., 4/8/32/128) or vs. pa buckets to quantify “higher pa → stronger shortening.”
3. **Sparse baselines under matched budgets**: No systematic comparisons against explicit length‑penalty PPO, length‑aware decoding/posterior selection, long‑to‑short distillation/compression, or GRPO variants with adaptive KL, all under equal token/compute budgets.
4. **Incomplete ablations**: λ vs. γ sweeps, KL schedules, and reward scaling effects are not systematically explored; claims about λ<1 stability would be stronger with controlled studies.  In unsolvable or all‑correct batches, the advantage vanishes and training can be driven by KL toward very short outputs; this is a collapse of learning signal rather than a controlled conciseness mechanism with accuracy guarantees.

**Questions:**

1.  **Full Phase‑1→Phase‑2 Demonstration**: Can a complete two-phase training curve (Phase‑1 on hard/low‑`pa` problems, Phase‑2 on "occasionally solvable" problems) be provided, showing the hypothesized length increase followed by a decrease, including results from at least 3-5 random seeds and error bars, to validate the full closed-loop mechanism?
2.  **Data Scale and Difficulty Distribution**: How do the Phase‑2 results vary with the number of training samples (e.g., 4/8/32/128) and across different `pa` difficulty buckets? Can the prediction that "higher `pa` → stronger shortening" be quantitatively verified?
3.  **Baseline Comparisons and Ablations**: Under matched compute/token budgets, how does Phase‑2 PPO compare against existing methods (e.g., PPO with explicit length penalties, length-aware decoding strategies, long-to-short distillation/compression, and stronger GRPO variants)? Furthermore, do ablation studies on key parameters (e.g., $\lambda$ and KL weight) clarify their impact on length and accuracy?
4.  **Statistical Robustness and Generalization**: Please provide multi-seed results, confidence intervals, and hypothesis tests for key benchmarks. Beyond math/STEM, can the method's generalization ability be demonstrated on a non-mathematical explanatory reasoning task?
5.  **Connecting Theory to Phenomena**: Can diagnostic data, such as the distribution of advantage signs over time, the evolution of termination token probabilities, and the relationship between value error and sequence length, be provided to more directly connect the theoretical analysis with observed training dynamics?

---

### Official Review · Reviewer_z3Bd · 2025-11-01

**Soundness:** 3
**Presentation:** 3
**Contribution:** 3
**Rating:** 8
**Confidence:** 2

**Summary:**

This paper tackles the problem of excessive verbosity in LLMs. The authors observe that state-of-the-art reasoning models often produce unnecessarily long chains of thought, incurring high computational cost and latency.

**Strengths:**

1. This work provides a fresh perspective by identifying verbosity as an optimization artifact rather than a necessity for reasoning. The theoretical analysis of PPO’s loss (Thm. 1) elegantly explains why incorrect answers lead to longer outputs. This analytical link between reward signals and sequence length is **novel and important**.
2. The experiments are thorough and convincing. The authors evaluate a mix of mathematics problem sets and a subset of MMLU covering science/engineering, providing evidence that the conciseness effect generalizes across domains (at least within STEM). They measure both chain-of-thought length and task accuracy, showing the desired trade-off clearly.
3. This work addresses a practical and impactful problem: the token inefficiency of current reasoning models. By showing that much of the verbosity can be cut without hurting accuracy, the paper has direct implications for deploying LLMs more efficiently (faster response times, lower API costs, less memory usage).
4. The proposed solution – a two-phase RL fine-tuning – is both effective and lightweight. It does not require collecting new human preference data or extensive annotations; instead, it leverages the model’s own correctness signal on a tiny set of problems. This is a compelling alternative to standard RLHF in scenarios where conciseness is desired.

**Weaknesses:**

1. Table 2 results are a little suspicious. For example, from Table 2, we do observe that this method has fewer tokens with higher accuracy; the improvement of accuracy seems marginal (57.0 to 58.4, and 71.6 to 72.1). The authors might want to report an error bar (standard deviation with 3+ runs on different seeds).
2. The possible reward hacking problem is under-discussed in this paper. During the training, does the model tend to have shorter responses to achieve a higher score? Can the reasoning chain really provide the answer?
3. Another minor limitation is that the evaluation focuses primarily on math and STEM reasoning tasks. While these are natural scenarios for testing chain-of-thought optimization (with well-defined correctness), it remains unclear how the approach would perform on other domains of reasoning. For instance, tasks like commonsense reasoning, legal or medical QA, or multi-hop fact reasoning might not have a binary “correct answer” reward signal.
3. (This is also a tiny concern that does not affect my decision and reviews, as I do acknowledge the difficulties in requiring GPU resources for 70B model training.) The experiments were conducted on relatively small models by today’s standards (1.5B and 7B parameters). It’s uncertain how a two-phase RL regimen would scale to much larger LLMs (e.g., 70B or 175B parameters)

**Questions:**

Questions:
1. Can this two-phase RL be applied to non-quantitative tasks like commonsense QA or dialogue,  where “correctness” is ill-defined? How would rewards be designed then?
2. How sensitive are the conclusions to this assumption? For tasks with intermediate rewards (e.g., multi-turn dialogue or program synthesis), would the PPO length-bias derivation still hold?
3. The two-phase approach uses extra GPU hours for a relatively small gain in accuracy but big length reduction. How should practitioners balance cost vs inference-time savings?

Suggestion:
1. While I understand this paper is more theoretical, adding some real model conversation will make the paper more solid. With the same question, for example, math500, what is the 1.5B R1 response and this method's response? Are these i

---

### Note · Authors · 2025-11-28

I have read and agree with the venue's withdrawal policy on behalf of myself and my co-authors.